Sterol regulatory element binding transcription factor 1 promotes proliferation and migration in head and neck squamous cell carcinoma

Tan Ming
Lin Xiaoyu
Chen Huiying
Ye Wanli
Yi Jianqi
Li Chao
Liu Jinlan
Su Jiping ymsu2@126.com
Department of Otolaryngology-Head and Neck Surgery, First Affiliated Hospital of Guangxi Medical University , Nanning, Guangxi , China
Tyagi Abhishek
Electronic publication date: 2023 Apr 17
Publication date: 2023
Volume: 11
Electronic Location ID: e15203
Received 2023 Jan 10; Accepted 2023 Mar 17
Copyright: ©2023 Tan et al.
Copyright year: 2023
Copyright holder: Tan et al.
License: This is an open access article distributed under the terms of the Creative Commons Attribution License, which permits unrestricted use, distribution, reproduction and adaptation in any medium and for any purpose provided that it is properly attributed. For attribution, the original author(s), title, publication source (PeerJ) and either DOI or URL of the article must be cited.
License URL: https://creativecommons.org/licenses/by/4.0/

Keywords: SREBF1, HNSC, Cell proliferation, Migration, Immune infiltration

Funding: The authors received no funding for this work.

==============================
Background

Sterol-regulatory element-binding protein 1 (SREBP1) is a transcription factor involved in lipid metabolism that is encoded by sterol regulatory element binding transcription factor 1(SREBF1). SREBP1 overexpression is associated with the progression of several human tumors; however, the role of SREBP1 in head and neck squamous cell carcinoma (HNSC) remains unclear.

Methods

SREBF1 expression in pan-cancer was analyzed using the Cancer Genome Atlas (TCGA) and Genotype-Tissue Expression (GTEx) data, and the association between SREBF1 expression and clinical characteristics of HNSC patients was examined using the UALCAN database. Enrichment analysis of SREBF1-related genes was performed using the Cluster Profiler R package. TCGA database was used to investigate the relationship between immune cell infiltration and SREBF1 expression. CCK-8, flow cytometry, and wound healing assays were performed to investigate the effect of SREBF1 knockdown on the proliferation and migration of HNSC cells.

Results

SREBF1 was significantly upregulated in several tumor tissues, including HNSC, and SREBF1 overexpression was positively correlated with sample type, cancer stage, tumor grade, and lymph node stage in HNSC patients. Gene enrichment analysis revealed that SREBF1 is associated with DNA replication and homologous recombination. SREBF1 upregulation was positively correlated with the infiltration of cytotoxic cells, B cells, T cells, T helper cells, and NK CD56 bright cells in HNSC. Knockdown of SREBF1 inhibited the proliferation and migration of HNSC cells (Hep2 and TU212) and induced apoptosis by downregulating the expression of steroidogenic acute regulatory protein-related lipid transfer 4 (STARD4).

Conclusions

SREBF1 may promote HNSC proliferation, migration and inhibit apoptosis by upregulating STARD4 and affecting the level of immune cell infiltration.

Introduction

Head and neck squamous cell carcinoma (HNSC) is the seventh most common malignancy worldwide and one of the most aggressive tumors (Ahmad Kiadaliri et al., 2013; Torre et al., 2015). More than 830,000 cases of HNSC are diagnosed worldwide each year, and over 430,000 people die from the disease (Cramer et al., 2019). Local recurrence and distant metastasis are the main causes of death in patients with HNSC. Although the diagnosis and treatment of HNSC have made progress in the past decades, the 5-year overall survival rates have not improved significantly (Yang et al., 2019). Despite more and more oncogenes having been identified with the continuous progress of transcriptome research and high-throughput sequencing technology, effective molecular biomarkers for detecting early HNSC and monitoring disease progression are lacking.

Growing evidence suggests that lipid metabolism reprogramming is ubiquitous in tumor cells (Schulze & Harris, 2012). Abnormal lipid metabolism promotes the malignant biological behavior of tumors (Luo et al., 2017; Tudek et al., 2017). In tumor cells, glucose and glutamine contribute to the synthesis of lipids in response to the PI3K/Akt (phosphatidylinositol-3-kinase/protein kinase B) signaling pathway and a series of key enzymes (Cheng et al., 2018). Sterol-regulatory element-binding protein 1 (SREBP1), encoded by the sterol regulatory element binding transcription factor 1 (SREBF1) gene, is an important nuclear transcription factor involved in lipid synthesis. SREBP1 is synthesized into inactive precursors on the endoplasmic reticulum and then transported to the Golgi apparatus where it is activated by proteases. Mature SREBP1 promotes lipid synthesis by activating the expression of downstream target genes (Han et al., 2015). Abnormal expression of SREBP1 is correlated with tumor progression in differentiated thyroid cancer, esophageal squamous cell carcinoma, glioblastoma, and ovarian cancer (Cheng et al., 2015; Huang et al., 2019; Koizume et al., 2019; Li et al., 2020b). SREBP1 is regulated by the PI3K/Akt oncogenic signaling pathway (Yi et al., 2020), which is activated in more than 90% of HNSC (Marquard & Jucker, 2020). Tumor immune cell infiltration plays an important role in tumor recurrence, metastasis, and immunotherapy (Jiang et al., 2018; Zeng et al., 2018). Immune cells are an important part of the tumor microenvironment in HNSC (Puram et al., 2017). However, there are no studies investigating the role of SREBF1 in the proliferation and immune infiltration of HNSC.

In this study, we will perform a comprehensive analysis of SREBF1 expression using multiple publicly available gene expression databases and investigate the correlation of SREBF1 expression in HNSC with sample type, cancer stage, lymph node status, and tumor grade. In addition, we will validate the expression of SREBF1 in HNSC and further investigate the effect of the knockdown of SREBF1 on the proliferation and migration of HNSC. This study provides a new idea for the targeting of SREBF1 in HNSC.

Materials and Methods

Gene expression analysis data

We combined The Cancer Genome Atlas (TCGA) and Genotype-Tissue Expression (GTEx) databases to analyze the differential expression of SREBF1 in pan-cancer and then analyzed the expression of SREBF1 and steroidogenic acute regulatory protein-related lipid transfer 4 (STARD4) in HNSC in TCGA paired and unpaired samples. The UALCAN database was then used to analyze the correlation of SREBF1 and STARD4 expression with sample type, tumor stage, lymph node stage, and tumor grade.

Correlation and enrichment analysis

We performed an enrichment analysis of genes significantly associated with SREBF1 expression in HNSC according to the TCGA database to understand the function of SREBF1. For enrichment analysis, the Cluster Profiler package in R software (version 3.6.3) was used.

Immune cell infiltration analysis

We used 24 kinds of immune cell markers to distinguish different immune cells according to a previous study (Bindea et al., 2013), and the correlation between immune cell infiltration and SREBF1 expression was analyzed using single sample gene enrichment analysis (ssGSEA) (Hanzelmann, Castelo & Guinney, 2013).

Cell culture and transfection

The Human HNSC cell lines Hep2 was donated by Professor Zhang zhe from the Nasopharyngeal Cancer Laboratory of Guangxi Medical University, and the SAS and SCC-9 were donated by Prof. Li Ping from Guangxi Medical University. TU212 was purchased from Beijing Zhongkezhijian Biotechnology Co., Ltd. FaDu and the normal human HOK were purchased from Shanghai WHELAB Bioscience Co., Ltd. Hep2 cells were cultured in DMEM with high glucose containing 10% fetal bovine serum (FBS, 10091148; Gibco, Billings, MT, USA) and 1% streptomycin and penicillin (P1400; Sigma Aldrich, St. Louis, MO, USA). The remaining cells were maintained according to the manufacturer’s protocol, and all cells were incubated at 37 °C, with 5% CO2. SREBF1-siRNA (Sense 5′-GCCUGACCAUCUGUGAGAATT-3′, antisense 5′-UUCUCACAGAUGGUCAGGCTT3′), Negative control SREBF1 - siNC (sense 5′-UUCUCCGAACGUGUCACGUTT-3′, antisense 5′-ACGUGACACGUUCGGAGAATT-3′) and GP-transfect-Mate reagent were purchased from Shanghai GenePharma Co., Ltd (G04008; Shanghai GenePharma Co. Ltd., Shanghai, China). Hep2 and TU212 cells were inoculated on 6-well plates, the OPTI-MEM (Cat. No. 31985070; Thermo Fisher Scientific, Waltham, MA, USA) mixture with 7.5 µl of GP-transfect-Mate transfection reagent and the OPTI-MEM mixture with 8.5 µl of SREBF1-siRNA/SREBF1-siNC were mixed well. Transfection was carried out after 15 min at room temperature. The transfected cells were used for the next experiment.

RNA Extraction and Real-Time RT-PCR(RT-qPCR)

Total RNA was extracted from cultured cells using TRIzol reagent (15596026; Invitrogen, Waltham, MA, USA), and reverse transcription was performed using a reverse transcription kit (AT311; TransGen Biotech, China). Reverse transcription-quantitative (RT-q) PCR was performed using the SYBR Green PCR kit (Invitrogen, USA, A25741). SREBF1 primers: forward 5′-ACAGTGACTTCCCTGGCCTAT-3′ and reverse 5′-GCATGGACGGGTACATCTTCAA-3′. STARD4 primers: forward 5′-TCCCTGTGGTTGGTTTTGTGTTCC-3′ and reverse 5′-TGGCTGTATCTACCGCAGACTGAG-3′. GAPDH primers: forward 5′-CAGGAGGCATTGCTGATGAT-3 ’and reverse 5′ -GAAGGCTGGGGCTCATTT-3′. GAPDH was used as an internal reference. The expression levels of mRNA were analyzed by the 2−ΔΔCT method. Each experiment was repeated at least three times.

Western blotting

Total cellular proteins were extracted from cultured cells (1.2 × 106/well) using cell lysate (RIPA buffer, protease inhibitor, and phosphatase inhibitor, P0013B), and the extracted proteins were mixed with 5 × SDS-PAGE protein loading solution at a ratio of 4:1 and then heated to degeneration sufficiently. After running the protein samples (80 µg) on SDS-PAGE gels, they were transferred to PVDF membranes, closed with 5% skim milk, and then mixed with the primary antibodies indicated (SREBF1 mouse monoclonal antibody (1:1000, Cat No. 66875-1-Ig; Proteintech, Wuhan, China), STARD4 antibody (1:1000, ab202060), GAPDH antibody (1:20000, Cat No. 60004-1-Ig; Proteintech, Wuhan, China) overnight at 4 °C, and then the PVDF membrane was scanned using infrared scanning equipment after incubation with secondary antibodies (Anti-mouse IgG (H+L), 1:15000, CST, 5470), Anti-rabbit IgG (H+L), 1:15000, CST, 5366). ImageJ software was used to analyze the relative expression of target proteins in membranes.

CCK-8 (Cell counting kit-8) assay

For this study, CCK-8 assays (Dojindo Cell counting tool test—8th Edition, Japan, JE603) were used. The transfected cells were plated in 96-well plates at a concentration of 3,000 cells per well. 10 µL CCK-8 solution was added to each well at 24 h, 48 h, 72 h, and 96 h, then the cells were incubated at 37 °C for 3 h. A 450-nm absorbance wavelength was used to measure the OD value of each well.

Wound healing assay

The transfected cells were plated in a 6-well plate with ibidi Culture-Insert, and incubated in the incubator for 24 h. After the Culture-Insert was removed, the cells were washed with PBS and incubated with 2% FBS. The cell migration status was recorded at 0 h and 24 h with the optical microscope.

Flow cytometric assessment

Cells were collected 24 h after transfection, washed twice with pre-chilled 1 × PBS, and resuspended to a density of approximately 1 × 106 cells/ml. 100 µL of the cell suspension was transferred to a 1.5 ml EP tube, and 5 µL of FITC Annexin V, and 5 µL of PI reagent (556547; BD Biosciences, San Diego, CA, USA) were added and mixed well. Incubate for 15 min at room temperature in the dark. Flow cytometry was then performed.

Statistical analysis

Data were expressed as the mean ± standard deviation (SD) and analyzed using SPSS 26.0 (SPSS, Chicago, IL, USA). The Shapiro–Wilk test was used to check the normality of the data. T test was used for samples that met the normality test, and the Wilcoxon rank sum test was used for samples that did not meet the normality test. P < 0.05 was considered statistically significant.

Results

Expression analysis of SREBF1 in pan-cancer

Analysis of the TCGA and GTEx databases revealed that SREBF1 expression is significantly increased in bladder urothelial carcinoma (BLCA) (P < 0.01), breast invasive carcinoma (BRCA), lymphoid neoplasm diffuse large B-cell lymphoma (DLBC), esophageal carcinoma (ESCA), HNSC, kidney chromophobe (KICH), kidney renal clear cell carcinoma (KIRC), kidney renal papillary cell carcinoma (KIRP), brain lower grade glioma (LGG), pancreatic adenocarcinoma (PAAD), stomach adenocarcinoma (STAD), and thymoma (THYM) (P < 0.001) tumors compared with adjacent normal tissues (Fig. 1A). Further analysis of the TCGA database revealed that SREBF1 and STARD4 expression is significantly higher in HNSC tumor tissues than in adjacent normal tissues (Figs. 1B–1E).

Figure 1 SREBF1 expression in human pan-cancers.

(A) TCGA and GTEx databases provide information on SREBF1 expression in tumors and adjacent normal tissues, HNSC (Normal = 44, Tumor = 520). (B–C) Expression of SREBF1 and STARD4 in adjacent normal tissues and tumors in HNSC(Normal = 44, Tumor = 502) from unpaired samples in TCGA. (D–E) Expression of SREBF1and STARD4 in tumor and adjacent normal tissues in HNSC (Normal = 43, Tumor = 43) from paired samples in TCGA. TPM (transcripts per million reads), Data are shown as the mean ± SD. *P < 0.05, **P < 0.01, ***P < 0.001.

SREBF1 expression and HNSC clinicopathology

Analysis of the UALCAN database indicated that the expression of SREBF1 and STARD4 correlated with sample type, tumor stage, lymph node stage, and tumor grade. As shown in Figs. 2A+2B, the expressions of SREBF1 and STARD4 were significantly higher in HNSC tumor tissues than in adjacent normal tissues (P < 0.001). Figures 2C+2D shows that the expressions of SREBF1 and STARD4 were significantly higher in mid to late-stage than in early-stage tumors. During cancer progression, the expressions of SREBF1 and STARD4 were significantly higher in the lymph node metastatic stage than in normal tissues (Figs. 2E+2F). Analyses of SREBF1 and STARD4 expressions with tumor grade indicated that their expressions increased significantly with increasing tumor grade (Figs. 2G+2H). These results suggest that SREBF1 and STARD4 play important roles in the pathogenesis of HNSC.

Figure 2 Correlation of SREBF1 and STARD4 mRNA expression with clinicopathological parameters in HNSC patients from the UALCAN database.

(A+B) Type of sample (normal/primary tumor). (C+D) Cancer stage (stage 1, 2, 3, and 4). (E+F) Lymph node stage (N0, 1, 2, and 3). (G+H) Tumor grade (Grades 1, 2, 3, and 4). N, normal; P, primary tumor; S1, stage 1; S2, stage 2; S3, stage 3; S4, stage 4.G1, Grade1; G2, Grade2; G3, Grade3; G4, Grade4.

Enrichment analysis of SREBF1-related genes

To investigate the potential mechanism of SREBF1 in the development of HNSC, we used the TCGA database screening SREBF1 expression-related mRNAs in HNSC (according to a study in the past p_spearman < 0.001 and — cor_spearman —> 0.4 (Chen et al., 2020)), then analyzed the potential function of these genes. The heat map shows the top 50 genes (Fig. 3), and the Kyoto Encyclopedia of Genes and Genomes (KEGG) enrichment analysis revealed that SREBF1 expression was associated with Fanconi anemia pathway, DNA replication, and homologous recombination (Fig. 4A). Gene Ontology (GO) analysis revealed that the three most enriched terms in biological process ontology were DNA replication, chromosome segregation, and DNA conformation change. In the cellular component ontology, the three most enriched terms were chromosomal region, nuclear chromatin, and spindle. For molecular functional ontology, the top three terms were catalytic activity, acting on DNA, histone binding, and single-stranded DNA binding (Figs. 4B–4D). These results indicate that overexpression of SREBF1 is associated with cell proliferation and that overexpression of SREBF1 in HNSC is associated with tumor progression.

Figure 3 Analysis of SREBF1-related genes.

The top 50 genes positively associated with SREBF1 expression are shown in the heat map. The data were normalized by the Z-score normalization method.

Figure 4 Enrichment analysis of SREBF1-related genes in HNSC.

(A) KEGG pathways of genes significantly associated with SREBF1. (B–D) Gene ontology terms are significantly associated with SREBF1 [including biological processes (B), cell components (C), and molecular function (D)].

SREBF1 expression is associated with immune cell infiltration

Immune cells that infiltrate tumors play a critical role in cancer progression (Fridman et al., 2011; Wu & Dai, 2017). We evaluated the correlation between SREBF1 expression in HNSC and the infiltration of 24 immune cells. The results showed that infiltration of NK CD56 bright cells and T helper cells was higher in the high SREBF1 expression group than in the low SREBF1 expression group (Fig. 5A). The expression of SREBF1 was significantly and positively correlated with the infiltration of NK CD56 bright cells and T helper cells (Fig. 5B). This suggested that high expression of SREBF1 in HNSC caused increased enrichment of NK CD56 bright cells and T helper cells, indicating that overexpression of SREBF1 was associated with immune activation in HNSC.

Figure 5 SREBF1 expression and immune cell infiltration in HNSC are correlated.

(A) Comparison of immune cell infiltration levels between SREBF1 differentially expressed groups in TCGA cohort of HNSC. (B) Correlation between SREBF1 and immune cell infiltration levels; red represents a positive correlation, green represents a negative correlation, and color shades represent the strength of the correlation. The data are presented as the mean ± standard deviation. *P < 0.05, **P < 0.01, ***P < 0.001. ns, not significant.

Knockdown of SREBF1 inhibits the proliferation and motility of HNSC cells

To assess the mechanism underlying the role of SREBF1 in HNSC progression, the target genes downstream of SREBF1 were first queried in the Chip-Atlas database (https://chip-atlas.org/). STARD4 is an important cholesterol transporter protein involved in the regulation of intracellular cholesterol homeostasis. Intracellular STARD4 binds free cholesterol to promote the formation of cholesteryl esters (Rodriguez-Agudo et al., 2008). Moreover, analysis in the TCGA database revealed a significant positive correlation between STARD4 and SREBF1 in HNSC (r = 0.239, P < 0.001, Fig. S1). Cellular assays showed that the expression of SREBF1 and STARD4 was significantly higher in HNSC cell lines than in normal human HOK cells (Fig. 6A). We then knocked down the expression of endogenous SREBF1 in HNSC cells (Hep2 and TU212) by RNA interference, RT-qPCR detected that SREBF1 was down-regulated along with STARD4 (Fig. 6B). The results of the Western blot assay were consistent with the results of the RT-qPCR assay (Fig. 6C). Next, we assessed cell proliferation using the CCK8 assay and found that the knockdown of SREBF1 significantly reduced the proliferation ability of HNSC cells (Hep2 and TU212) (Figs. 6D–6E). Knockdown of SREBF1 significantly increased apoptosis in HNSC cells (Hep2 and TU212) as shown by flow cytometry (Figs. 6F–6G). Silencing of SREBF1 significantly decreased the migration ability of HNSC cells (Hep2 and TU212) as shown by wound healing assays (Fig. 6H). Taken together, these results suggested that SREBF1 may promote the proliferation and migration of HNSC through the upregulation of STARD4.

Figure 6 Knockdown of SREBF1 inhibited the proliferation and migration of HNSC cells.

(A) Expression levels of SREBF1 mRNA and STARD4 mRNA were measured in HNSC cells by RT-qPCR. (B) SREBF1 was knocked down by RNA interference, and the levels of SREBF1 and STARD were assessed by RT-qPCR. (C) Western blot analysis of SREBF1 and STARD4 levels. (D–G) CCK-8 assay and flow cytometric assessment were used to detect HNSC cell growth. (H) A wound-healing assay was used to detect the capacity of HNSC cells to migrate (original magnification × 100). The data are presented as the mean ± SD. *P < 0.05, **P < 0.01, ***P < 0.001, ****P < 0.0001.

Discussion

The five-year overall survival rate of HNSC has been hovering around 50% (Thariat et al., 2015), so it is urgent to further understand the pathogenesis of this kind of disease. Lipid metabolism in malignant tumors has become the focus of research. Reprogramming of lipid metabolism is one of the hallmarks of malignancy (Cheng et al., 2018). Lipids not only serve as important components of biological membranes but also play a critical role in cellular signal transduction processes (Rohrig & Schulze, 2016). SREBP1 is a key transcription factor involved in the regulation of lipid metabolism, and it plays a regulatory role in various human metabolic diseases; it also serves as a hub linking oncogenic signaling and tumor metabolism (Guo et al., 2014). SREBP1 expression is significantly increased in tumors and plays an integral role in tumor progression (Gao et al., 2019; Zhou et al., 2019; Zhou et al., 2020). SREBP1 expression is significantly higher in breast cancer than in adjacent normal tissues, and knockdown of SREBP1 inhibits the proliferation, migration, and invasion of tumor cells (Zhang et al., 2019). SREBP1 is overexpressed in renal clear cell carcinoma, and silencing SREBP1 inhibits tumor progression through the NF-κB signaling pathway (Yang et al., 2018). Knockdown of SREBP1 in colon cancer inhibits tumor growth by altering cellular metabolism through the downregulation of genes related to lipid metabolism (Wen et al., 2018). Upregulation of SREBP1 expression in human hepatocellular carcinoma correlates with a poor prognosis of patients (Li et al., 2014). This suggests that SREBP1 plays an oncogenic role in the progression of several human cancers. Novel inhibitors of SREBP1 can significantly inhibit the growth of hepatocellular carcinoma and prostate cancer (Meng et al., 2021; Singh et al., 2019). These results suggest that SREBP1 is a novel molecular target in cancer. However, the expression of SREBF1 in human cancers and its potential mechanism of action remain unclear. In the present study, we used the TCGA and GTEx databases to analyze SREBF1 expression in human pan-cancer and found that SREBF1 was significantly upregulated in 12 cancers, including BLCA, BRCA, DLBC, ESCA, HNSC, KICH, KIRC, KIRP, LGG, PAAD, STAD, and THYM. However, the expression of SREBF1 was reduced in other tumors, such as adrenocortical carcinoma (ACC), colon adenocarcinoma (COAD), glioblastoma multiforme (GBM), acute myeloid leukemia (LAML), liver hepatocellular carcinoma (LIHC), lung adenocarcinoma (LUAD), ovarian serous cystadenocarcinoma (OV), pheochromocytoma and paraganglioma (PCPG), rectum adenocarcinoma (READ), skin cutaneous melanoma (SKCM), testicular germ cell tumors (TGCT), thyroid carcinoma (THCA), and uterine carcinosarcoma (UCS). This suggests that SREBF1 plays an important role in the progression of many tumors, the role of SREBF1 in low-expressing tumors may need to be investigated in depth. Previous studies have shown that Fatostatin, an inhibitor of SREBP1, can reduce the activity of cancer cells (HeLa, SH-SY5Y, and U2OS) and normal cells (RPE and MEFs), but has a more pronounced effect on cancer cell viability (∼10–20% cell viability vs ∼50–62% cell viability) (Gholkar et al., 2016; Ma et al., 2021). These findings suggest that the use of SREBP1 inhibitors in the targeted treatment of cancers with high SREBP1 expression may also interfere with the normal tissues, resulting in damage to normal cells and other side effects. Therefore, further toxicological experiments are needed to objectively evaluate the application of targeted therapy in tumors with high SREBF1 expression.

Furthermore, analysis of the TCGA and UALCAN databases revealed that the expression of SREBF1 was significantly higher in HNSC than in normal tissues. In addition, high expression of SREBF1 correlated with tumor stage, lymph node stage, and tumor grade in HNSC, suggesting that SREBF1 plays an important role in the progression of HNSC.

To further investigate the role of SREBF1 in HNSC, enrichment analysis of related genes showed that SREBF1 is associated with cell proliferation pathways such as DNA replication and homologous recombination.

Further biological experiments indicated that the expression levels of SREBF1 and STARD4 were significantly increased in HNSC cells. Knockdown of SREBF1 significantly inhibited the proliferation and migration of HNSC cell lines (Hep2 and TU212). The potential mechanisms underlying the role of SREBP1 in tumor progression include promoting tumor growth by increasing lipid synthesis through the activation of its target genes, as shown in prostate cancer, gastric cancer, and bladder cancer (Du et al., 2012; Miyachi et al., 2013; Singh et al., 2019). In the present study, the knockdown of SREBF1 in HNSC cells (Hep2 and TU212) downregulated the expression of STARD4. High expression of STARD4 in breast cancer has also been reported, and cell function experiments show that knockdown of STARD4 significantly inhibits the proliferation and migration of breast cancer. This suggests that SREBF1 may promote the proliferation and migration of head and neck squamous cell carcinoma through STARD4. In this study, we showed that SREBF1 is associated with the Fanconi anemia pathway, which is associated with an increased risk of HNSC (Vigneswaran & Williams, 2014). This supports the potential association of SREBF1 with the development of HNSC and provides a new theoretical basis for understanding the role of SREBF1 in promoting the progression of HNSC.

Immune cells are an important component of the tumor microenvironment and play an important role in regulating the malignant behavior of tumor cells (Binnewies et al., 2018; Sahin Ozkan et al., 2020; Vilarino et al., 2020). Immune cell infiltration in the tumor microenvironment is an important predictor of prognosis and treatment outcome in cancer patients (Lee et al., 2017; Li et al., 2020a). Recent studies show that tumor immune cell infiltration is associated with the prognosis of HNSC (Zhang et al., 2020). However, whether SREBF1 expression is associated with immune cell infiltration in HNSC remains unclear. We comprehensively analyzed the correlation between SREBF1 expression and the level of immune cell infiltration in HNSC. The results showed that the infiltration of T helper cells and NK CD56 bright cells was higher in the HNSC group with high SREBF1 expression than in the low expression group. Moreover, the expression of SREBF1 was significantly and positively correlated with the infiltration level of T helper cells and NK CD56 bright cells. Metabolic reprogramming supports the production of IFN-γ by NK CD56 bright cells in the immune response (Keating et al., 2016), and IFN-γ has immunomodulatory and antitumor effects. T helper cells are also associated with immune responses in the tumor environment (Ostroumov et al., 2018). This suggests that SREBF1 is involved in regulating the HNSC immune response.

Conclusion

In conclusion, SREBF1 possibly through upregulation of STARD4 and affects immune infiltration to promote proliferation, migration and inhibit apoptosis in head and neck squamous cell carcinoma.

Supplemental Information

Figure S1 Correlation analysis of SREBF1 and STARD4 in the HNSC cohort of TCGA database

Click here for additional data file.

Additional Information and Declarations

Competing Interests

Author Contributions

Data Availability

The authors declare there are no competing interests.

Ming Tan conceived and designed the experiments, authored or reviewed drafts of the article, and approved the final draft.

Xiaoyu Lin conceived and designed the experiments, analyzed the data, authored or reviewed drafts of the article, and approved the final draft.

Huiying Chen conceived and designed the experiments, prepared figures and/or tables, and approved the final draft.

Wanli Ye performed the experiments, analyzed the data, prepared figures and/or tables, and approved the final draft.

Jianqi Yi performed the experiments, prepared figures and/or tables, and approved the final draft.

Chao Li conceived and designed the experiments, performed the experiments, prepared figures and/or tables, and approved the final draft.

Jinlan Liu performed the experiments, prepared figures and/or tables, and approved the final draft.

Jiping Su performed the experiments, authored or reviewed drafts of the article, and approved the final draft.

The following information was supplied regarding data availability:

The data is available at figshare: Tan, Ming (2023): raw data 2022-12-30. figshare. Journal contribution. https://doi.org/10.6084/m9.figshare.21836973.v1.

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
