# Peer review of "Sterol regulatory element binding transcription factor 1 promotes proliferation and migration in head and neck squamous cell carcinoma"

_PeerJ, doi:10.7717/peerj.15203_

## Round 0.1 · original submission · Major Revisions

Dear Dr. Tan,

Thank you for submitting your manuscript "Overexpression of SREBF1 is associated with proliferation and immune infiltration of HNSC" to PeerJ. We have now received reports from the reviewers, and, after careful consideration internally, we have decided to invite a major revision of the manuscript.

As you will see from the reports copied below, the reviewers raise important concerns regarding the detailed methodology, the experimental validation/quantification of the results, and their interpretation and further clarification of the experimental design. We find that these concerns limit the strength of the study, and therefore we ask you to address them with additional work. Without substantial revisions, we will be unlikely to send the paper back for review.

If you feel that you are able to comprehensively address the reviewers’ concerns, please provide a point-by-point response to these comments along with your revision. Please show all changes in the manuscript text file with track changes or color highlighting. If you are unable to address specific reviewer requests or find any points invalid, please explain why in the point-by-point response.

Thanks

Abhishek Tyagi, PhD
Academic Editor,
PeerJ

·

Basic reporting

no comment

Experimental design

no comment

Validity of the findings

no comment

Additional comments

In this study, the comprehensive analysis found that SREBF1 was significantly higher in tumors, which was correlated with immune cell infiltration in tumors. Knockdown of SREBF1 inhibited the proliferation and migration of HNSC cells, revealing the oncogenic role for SREBF1 in HNSC and providing a new idea for the targeted therapy of SREBF1 in HNSC. However, some concerns need to be clarified.
1. the full spelling of SREBF1 was not presented in the manuscript, instead of repeated full spelling for SREBP1;
2. What’s the concentration of siRNA for SREBF1? Add the details for RNA interference assay;
3. The amounts of cells for protein extraction and the amount of protein for western blotting should be added. The details of GAPDH antibody and secondary antibody should be added;
4. The product code of all reagents should be added;
5. The data analysis and its description are insufficient;
6. How many samples were analysed for tumors and normal tissues from different cancers in Figure 1A-C?
7. In lines 219-221, “These results indicate that overexpression of SREBF1 is associated with cell proliferation and that overexpression of SREBF1 in HNSC is associated with tumor progression”, however, the cell proliferation pathway was not found in the KEGG and GO analysis of SREBF1-related genes. How to obtain the above conclusion?
8. In the discussion, the findings are similar for all cancer tissues, therefore, the new results from this experiment should be emphasized in the discussion, subtitle could be used to stress the focal points;
9. The conclusion of this study that SREBF1 may play an oncogenic role in HNSC was not fully supported by the results from overexpression and correlation analysis, additional experiment to prove the pathway of such regulation should be conducted;
10. References: Please keep the format consistent, especially for titles.

Reviewer 2 ·

Basic reporting

In this study, the authors focused on SREBF1 in HNSCs and analyzed its cancer relevance. Several results supporting the cancer relevance were obtained, including their own experiments. The structure and flow of the analysis seemed appropriate. However, there are several points that must be clarified:

1. Lines 76-83 in Introduction appear to be Abstract or Result. The purpose of the study should be stated.

2. Figure 1, 2. There are some garbled characters.

3. Figure 1~6. The resolution is poor. In particular, the experimental results in Figure 6 should be of higher resolution.

Experimental design

no comment

Validity of the findings

4. Figure 1. Comparison of tumors and adjacent normal tissues shows significant differences, but there are several normal tissues with higher SREBF1 expression than cancer, such as normal tissues of ACC, PCPG, and TGCT. What is the function of SREBF1 in these normal tissues? Considering the possibility of "the targeted therapy of SREBF1" as described in l.83, I am concerned about the effect/toxicity of SREBF1 on normal cells, which are more highly expressed than those in cancer.

5. Figure 6. The growth inhibitory effect of SREBF1-siRNA on multiple cells (Hep2 and TU212) seems reasonable. In addition to this, there are now many public databases of the effects of RNAi on proliferation. For example, DepMap (https://depmap.org/portal/gene/SREBF1). If possible, I would like to know whether this findings are supported by such RNAi databases.

---

## Round 0.2 · Minor Revisions

Dear Dr. Tan,

Thank you for your submission to PeerJ.
After assessing your revised manuscript, it needs minor revisions as recommended by the reviewer. I recommend addressing them.

Thanks

Abhishek Tyagi, PhD

Reviewer 2 ·

Basic reporting

The authors have improved the manuscript and have addressed most of my comments. I have one additional request and one comment.

Experimental design

no comment

Validity of the findings

Regarding point 4, I am pleased to see authors' discussion of SREBF1 low expressing tumors. My wording was inadequate. My point was not about the role of SREBF1 in low SREBF1-expressing tumors, but about the possibility that an inhibitor that targets SREBF1-high expressing tumors may also inhibit SREBF1-high expressing normal tissues and cause toxicity by disrupting the normal function of SREBF1 in normal tissues. Some additional analysis or discussion of the effects of SREBF1 inhibition on SREBF1-high normal tissues would be desirable.

Regarding point 5, it is regrettable that authors could not find any supporting information while large scale RNAi screen data is available on DepMap (The Cancer Dependency Map) and Project Achilles. I felt that the reproducibility of the findings may be limited.

---

## Round 0.3 · accepted · Accept

Dear Dr. Tan,

Your manuscript has been Accepted for publication.

Reviewer 2 ·

Basic reporting

The authors have fully addressed all of my comments with additional reference data and appropriate discussion. Therefore, I would like to recommend that it move to publication.

Experimental design

no comment

Validity of the findings

no comment